# A Deep Reinforced Model for Abstractive Summarization

**Romain Paulus, Caiming Xiong**∗**& Richard Socher**
Salesforce Research
575 High Street
Palo Alto, CA 94301, USA
{rpaulus,cxiong,rsocher}@salesforce.com

## Abstract

Attentional, RNN-based encoder-decoder models for abstractive summarization have achieved good performance on short input and output sequences. For longer documents and summaries however these models often include repetitive and incoherent phrases. We introduce a neural network model with a novel intra-attention that attends over the input and continuously generated output separately, and a new training method that combines standard supervised word prediction and reinforcement learning (RL). Models trained only with supervised learning often exhibit "exposure bias" – they assume ground truth is provided at each step during training. However, when standard word prediction is combined with the global sequence prediction training of RL the resulting summaries become more readable. We evaluate this model on the CNN/Daily Mail and New York Times datasets. Our model obtains a 41.16 ROUGE-1 score on the CNN/Daily Mail dataset, an improvement over previous state-of-the-art models. Human evaluation also shows that our model produces higher quality summaries.

## 1 Introduction

Text summarization is the process of automatically generating natural language summaries from an input document while retaining the important points. By condensing large quantities of information into short, informative summaries, summarization can aid many downstream applications such as creating news digests, search, and report generation.

There are two prominent types of summarization algorithms. First, extractive summarization systems form summaries by copying parts of the input (Dorr et al., 2003; Nallapati et al., 2017). Second, abstractive summarization systems generate new phrases, possibly rephrasing or using words that were not in the original text (Chopra et al., 2016; Nallapati et al., 2016).

Neural network models (Nallapati et al., 2016) based on the attentional encoder-decoder model for machine translation (Bahdanau et al., 2015) were able to generate abstractive summaries with high ROUGE scores. However, these systems have typically been used for summarizing short input sequences (one or two sentences) to generate even shorter summaries. For example, the summaries on the DUC-2004 dataset generated by the state-of-the-art system by Zeng et al. (2016) are limited to 75 characters.

Nallapati et al. (2016) also applied their abstractive summarization model on the CNN/Daily Mail dataset (Hermann et al., 2015), which contains input sequences of up to 800 tokens and multi-sentence summaries of up to 100 tokens. But their analysis illustrates a key problem with attentional encoder-decoder models: they often generate unnatural summaries consisting of repeated phrases.

We present a new abstractive summarization model that achieves state-of-the-art results on the CNN/Daily Mail and similarly good results on the New York Times dataset (NYT) (Sandhaus, 2008). To our knowledge, this is the first end-to-end model for abstractive summarization on the NYT dataset. We introduce a key attention mechanism and a new learning objective to address the

---

∗Corresponding author.

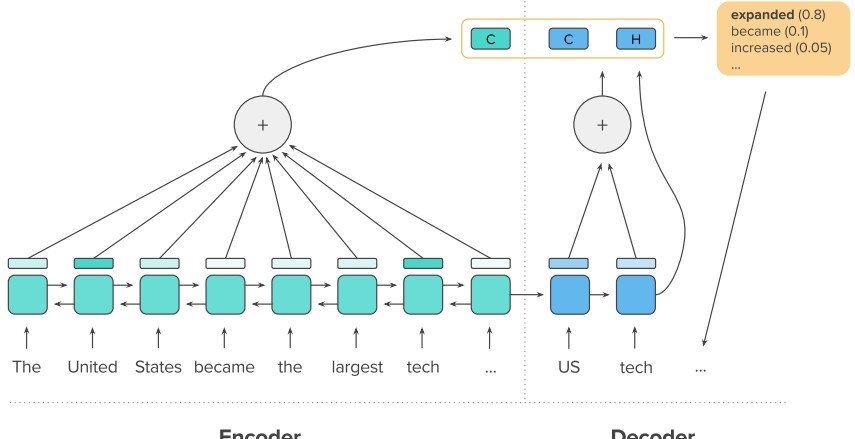

Figure 1: Illustration of the encoder and decoder attention functions combined. The two context vectors (marked "C") are computed from attending over the encoder hidden states and decoder hidden states. Using these two contexts and the current decoder hidden state ("H"), a new word is generated and added to the output sequence.

repeating phrase problem: (i) we use an intra-temporal attention in the encoder that records previous attention weights for each of the input tokens while a sequential intra-attention model in the decoder takes into account which words have already been generated by the decoder. (ii) we propose a new objective function by combining the maximum-likelihood cross-entropy loss used in prior work with rewards from policy gradient reinforcement learning to reduce exposure bias.

Our model achieves 41.16 ROUGE-1 on the CNN/Daily Mail dataset. Moreover, we show, through human evaluation of generated outputs, that our model generates more readable summaries compared to other abstractive approaches.

## 2 NEURAL INTRA-ATTENTION MODEL

In this section, we present our intra-attention model based on the encoder-decoder network (Sutskever et al., 2014). In all our equations, $x = \{x_1, x_2, \ldots, x_n\}$ represents the sequence of input (article) tokens, $y = \{y_1, y_2, \ldots, y_{n'}\}$ the sequence of output (summary) tokens, and $\parallel$ denotes the vector concatenation operator.

Our model reads the input sequence with a bi-directional LSTM encoder $\{\mathrm{RNN}^{e\_\mathrm{fwd}}, \mathrm{RNN}^{e\_\mathrm{bwd}}\}$ computing hidden states $h_i^e = [h_i^{e\_\mathrm{fwd}} \| h_i^{e\_\mathrm{bwd}}]$ from the embedding vectors of $x_i$. We use a single LSTM decoder $\mathrm{RNN}^d$, computing hidden states $h_t^d$ from the embedding vectors of $y_t$. Both input and output embeddings are taken from the same matrix $W_{\mathrm{emb}}$. We initialize the decoder hidden state with $h_0^d = h_n^e$.

### 2.1 INTRA-TEMPORAL ATTENTION ON INPUT SEQUENCE

At each decoding step $t$, we use an intra-temporal attention function to attend over specific parts of the encoded input sequence in addition to the decoder's own hidden state and the previously-generated word (Sankaran et al., 2016). This kind of attention prevents the model from attending over the sames parts of the input on different decoding steps. Nallapati et al. (2016) have shown that such an intra-temporal attention can reduce the amount of repetitions when attending over long documents.

We define $e_{ti}$ as the attention score of the hidden input state $h_i^e$ at decoding time step $t$:

$$e_{ti} = f(h_t^d, h_i^e), \tag{1}$$

where $f$ can be any function returning a scalar $e_{ti}$ from the $h_t^d$ and $h_i^e$ vectors. While some attention models use functions as simple as the dot-product between the two vectors, we choose to use a

bilinear function:

$$f(h_t^d, h_i^e) = h_t^{d^T} W_{\text{attn}}^e h_i^e. \tag{2}$$

We normalize the attention weights with the following temporal attention function, penalizing input tokens that have obtained high attention scores in past decoding steps. We define new temporal scores $e'_{ti}$:

$$e'_{ti} = \begin{cases} exp(e_{ti}) & \text{if } t = 1 \\ \frac{exp(e_{ti})}{\sum_{j=1}^{t-1} \exp(e_{ji})} & \text{otherwise.} \end{cases} \tag{3}$$

Finally, we compute the normalized attention scores $\alpha_{ti}^e$ across the inputs and use these weights to obtain the input context vector $c_t^e$:

$$\alpha_{ti}^e = \frac{e'_{ti}}{\sum_{j=1}^n e'_{tj}} \tag{4} \qquad\qquad c_t^e = \sum_{i=1}^n \alpha_{ti}^e h_i^e. \tag{5}$$

## 2.2 Intra-decoder attention

While this intra-temporal attention function ensures that different parts of the encoded input sequence are used, our decoder can still generate repeated phrases based on its own hidden states, especially when generating long sequences. To prevent that, we can incorporate more information about the previously decoded sequence into the decoder. Looking back at previous decoding steps will allow our model to make more structured predictions and avoid repeating the same information, even if that information was generated many steps away. To achieve this, we introduce an intra-decoder attention mechanism. This mechanism is not present in existing encoder-decoder models for abstractive summarization.

For each decoding step $t$, our model computes a new decoder context vector $c_t^d$. We set $c_1^d$ to a vector of zeros since the generated sequence is empty on the first decoding step. For $t > 1$, we use the following equations:

$$e_{tt'}^d = h_t^{d^T} W_{\text{attn}}^d h_{t'}^d \tag{6} \qquad \alpha_{tt'}^d = \frac{exp(e_{tt'}^d)}{\sum_{j=1}^{t-1} exp(e_{tj}^d)} \tag{7} \qquad c_t^d = \sum_{j=1}^{t-1} \alpha_{tj}^d h_j^d \tag{8}$$

Figure 1 illustrates the intra-attention context vector computation $c_t^d$, in addition to the encoder temporal attention, and their use in the decoder.

A closely-related intra-RNN attention function has been introduced by Cheng et al. (2016) but their implementation works by modifying the underlying LSTM function, and they do not apply it to long sequence generation problems. This is a major difference with our method, which makes no assumptions about the type of decoder RNN, thus is more simple and widely applicable to other types of recurrent networks.

## 2.3 Token generation and pointer

To generate a token, our decoder uses either a token-generation softmax layer or a pointer mechanism to copy rare or unseen from the input sequence. We use a switch function that decides at each decoding step whether to use the token generation or the pointer (Gulcehre et al., 2016; Nallapati et al., 2016). We define $u_t$ as a binary value, equal to 1 if the pointer mechanism is used to output $y_t$, and 0 otherwise. In the following equations, all probabilities are conditioned on $y_1, \ldots, y_{t-1}, x$, even when not explicitly stated.

Our token-generation layer generates the following probability distribution:

$$p(y_t | u_t = 0) = \text{softmax}(W_{\text{out}}[h_t^d \| c_t^e \| c_t^d] + b_{\text{out}}) \tag{9}$$

On the other hand, the pointer mechanism uses the temporal attention weights $\alpha_{ti}^e$ as the probability distribution to copy the input token $x_i$.

$$p(y_t = x_i | u_t = 1) = \alpha_{ti}^e \tag{10}$$

We also compute the probability of using the copy mechanism for the decoding step $t$:

$$p(u_t = 1) = \sigma(W_u[h_t^d \| c_t^e \| c_t^d] + b_u), \tag{11}$$

where $\sigma$ is the sigmoid activation function.

Putting Equations 9 , 10 and 11 together, we obtain our final probability distribution for the output token $y_t$:

$$p(y_t) = p(u_t = 1)p(y_t|u_t = 1) + p(u_t = 0)p(y_t|u_t = 0). \tag{12}$$

The ground-truth value for $u_t$ and the corresponding $i$ index of the target input token when $u_t = 1$ are provided at every decoding step during training. We set $u_t = 1$ either when $y_t$ is an out-of-vocabulary token or when it is a pre-defined named entity (see Section 5).

### 2.4 SHARING DECODER WEIGHTS

In addition to using the same embedding matrix $W_{\text{emb}}$ for the encoder and the decoder sequences, we introduce some weight-sharing between this embedding matrix and the $W_{\text{out}}$ matrix of the token-generation layer, similarly to Inan et al. (2017) and Press & Wolf (2016). This allows the token-generation function to use syntactic and semantic information contained in the embedding matrix.

$$W_{\text{out}} = \tanh(W_{\text{emb}}W_{\text{proj}}) \tag{13}$$

### 2.5 REPETITION AVOIDANCE AT TEST TIME

Another way to avoid repetitions comes from our observation that in both the CNN/Daily Mail and NYT datasets, ground-truth summaries almost never contain the same trigram twice. Based on this observation, we force our decoder to never output the same trigram more than once during testing. We do this by setting $p(y_t) = 0$ during beam search, when outputting $y_t$ would create a trigram that already exists in the previously decoded sequence of the current beam.

## 3 HYBRID LEARNING OBJECTIVE

In this section, we explore different ways of training our encoder-decoder model. In particular, we propose reinforcement learning-based algorithms and their application to our summarization task.

### 3.1 SUPERVISED LEARNING WITH TEACHER FORCING

The most widely used method to train a decoder RNN for sequence generation, called the teacher forcing" algorithm (Williams & Zipser, 1989), minimizes a maximum-likelihood loss at each decoding step. We define $y^* = \{y_1^*, y_2^*, \ldots, y_{n'}^*\}$ as the ground-truth output sequence for a given input sequence $x$. The maximum-likelihood training objective is the minimization of the following loss:

$$L_{ml} = -\sum_{t=1}^{n'} \log p(y_t^*|y_1^*, \ldots, y_{t-1}^*, x) \tag{14}$$

However, minimizing $L_{ml}$ does not always produce the best results on discrete evaluation metrics such as ROUGE (Lin, 2004). This phenomenon has been observed with similar sequence generation tasks like image captioning with CIDEr (Rennie et al., 2016) and machine translation with BLEU (Wu et al., 2016; Norouzi et al., 2016). There are two main reasons for this discrepancy. The first one, called exposure bias (Ranzato et al., 2015), comes from the fact that the network has knowledge of the ground truth sequence up to the next token during training but does not have such supervision when testing, hence accumulating errors as it predicts the sequence. The second reason is due to the large number of potentially valid summaries, since there are more ways to arrange tokens to produce paraphrases or different sentence orders. The ROUGE metrics take some of this flexibility into account, but the maximum-likelihood objective does not.

## 3.2 POLICY LEARNING

One way to remedy this is to learn a policy that maximizes a specific discrete metric instead of minimizing the maximum-likelihood loss, which is made possible with reinforcement learning. In our model, we use the self-critical policy gradient training algorithm (Rennie et al., 2016).

For this training algorithm, we produce two separate output sequences at each training iteration: $y^s$, which is obtained by sampling from the $p(y_t^s|y_1^s, \ldots, y_{t-1}^s, x)$ probability distribution at each decoding time step, and $\hat{y}$, the baseline output, obtained by maximizing the output probability distribution at each time step, essentially performing a greedy search. We define $r(y)$ as the reward function for an output sequence $y$, comparing it with the ground truth sequence $y^*$ with the evaluation metric of our choice.

$$L_{rl} = (r(\hat{y}) - r(y^s)) \sum_{t=1}^{n'} \log p(y_t^s|y_1^s, \ldots, y_{t-1}^s, x) \tag{15}$$

We can see that minimizing $L_{rl}$ is equivalent to maximizing the conditional likelihood of the sampled sequence $y^s$ if it obtains a higher reward than the baseline $\hat{y}$, thus increasing the reward expectation of our model.

## 3.3 MIXED TRAINING OBJECTIVE FUNCTION

One potential issue of this reinforcement training objective is that optimizing for a specific discrete metric like ROUGE does not guarantee an increase in quality and readability of the output. It is possible to game such discrete metrics and increase their score without an actual increase in readability or relevance (Liu et al., 2016). While ROUGE measures the n-gram overlap between our generated summary and a reference sequence, human-readability is better captured by a language model, which is usually measured by perplexity.

Since our maximum-likelihood training objective (Equation 14) is essentially a conditional language model, calculating the probability of a token $y_t$ based on the previously predicted sequence $\{y_1, \ldots, y_{t-1}\}$ and the input sequence $x$, we hypothesize that it can assist our policy learning algorithm to generate more natural summaries. This motivates us to define a mixed learning objective function that combines equations 14 and 15:

$$L_{mixed} = \gamma L_{rl} + (1 - \gamma)L_{ml}, \tag{16}$$

where $\gamma$ is a scaling factor accounting for the difference in magnitude between $L_{rl}$ and $L_{ml}$. A similar mixed-objective learning function has been used by Wu et al. (2016) for machine translation on short sequences, but this is its first use in combination with self-critical policy learning for long summarization to explicitly improve readability in addition to evaluation metrics.

## 4 RELATED WORK

### 4.1 NEURAL ENCODER-DECODER SEQUENCE MODELS

Neural encoder-decoder models are widely used in NLP applications such as machine translation (Sutskever et al., 2014), summarization (Chopra et al., 2016; Nallapati et al., 2016), and question answering (Hermann et al., 2015). These models use recurrent neural networks (RNN), such as long-short term memory network (LSTM) (Hochreiter & Schmidhuber, 1997) to encode an input sentence into a fixed vector, and create a new output sequence from that vector using another RNN. To apply this sequence-to-sequence approach to natural language, word embeddings (Mikolov et al., 2013; Pennington et al., 2014) are used to convert language tokens to vectors that can be used as inputs for these networks. Attention mechanisms (Bahdanau et al., 2015) make these models more performant and scalable, allowing them to look back at parts of the encoded input sequence while the output is generated. These models often use a fixed input and output vocabulary, which prevents them from learning representations for new words. One way to fix this is to allow the decoder network to point back to some specific words or sub-sequences of the input and copy them onto the output sequence (Vinyals et al., 2015). Gulcehre et al. (2016) and Merity et al. (2017) combine this pointer mechanism with the original word generation layer in the decoder to allow the model to use either method at each decoding step.

## 4.2 Reinforcement learning for sequence generation

Reinforcement learning (RL) is a way of training an agent to interact with a given environment in order to maximize a reward. RL has been used to solve a wide variety of problems, usually when an agent has to perform discrete actions before obtaining a reward, or when the metric to optimize is not differentiable and traditional supervised learning methods cannot be used. This is applicable to sequence generation tasks, because many of the metrics used to evaluate these tasks (like BLEU, ROUGE or METEOR) are not differentiable.

In order to optimize that metric directly, Ranzato et al. (2015) have applied the REINFORCE algorithm (Williams, 1992) to train various RNN-based models for sequence generation tasks, leading to significant improvements compared to previous supervised learning methods. Bahdanau et al. (2017) also use a different kind of reinforcement learning algorithm on machine to optimize BLEU scores in machine translation tasks. While both these methods require an additional neural network, called a critic model, to predict the expected reward and stabilize the objective function gradients, Rennie et al. (2016) designed a self-critical sequence training method that does not require this critic model and lead to further improvements on image captioning tasks.

## 4.3 Text summarization

Most summarization models studied in the past are extractive in nature (Dorr et al., 2003; Nallapati et al., 2017; Durrett et al., 2016), which usually work by identifying the most important phrases of an input document and re-arranging them into a new summary sequence. The more recent abstractive summarization models have more degrees of freedom and can create more novel sequences. Many abstractive models such as Rush et al. (2015), Chopra et al. (2016) and Nallapati et al. (2016) are all based on the neural encoder-decoder architecture (Section 4.1). Miao & Blunsom (2016) extend the encoder-decoder architecture with a variational auto-encoder, and use REINFORCE to train it as well.

A well-studied set of summarization tasks is the Document Understanding Conference (DUC) [1]. These summarization tasks are varied, including short summaries of a single document and long summaries of multiple documents categorized by subject. Most abstractive summarization models have been evaluated on the DUC-2004 dataset, and outperform extractive models on that task (Dorr et al., 2003). However, models trained on the DUC-2004 task can only generate very short summaries up to 75 characters, and are usually used with one or two input sentences. Chen et al. (2016) applied different kinds of attention mechanisms for summarization on the CNN dataset, and Nallapati et al. (2016) used different attention and pointer functions on the CNN and Daily Mail datasets combined. In parallel of our work, See et al. (2017) also developed an abstractive summarization model on this dataset with an extra loss term to increase temporal coverage of the encoder attention function.

## 5 Datasets

### 5.1 CNN/Daily Mail

We evaluate our model on a modified version of the CNN/Daily Mail dataset (Hermann et al., 2015), following the same pre-processing steps described in Nallapati et al. (2016). We refer the reader to that paper for a detailed description. The final dataset contains 287,113 training examples, 13,368 validation examples and 11,490 testing examples. After limiting the input length to 800 tokens and output length to 100 tokens, the average input and output lengths are respectively 632 and 53 tokens.

### 5.2 New York Times

The New York Times (NYT) dataset (Sandhaus, 2008) is a large collection of articles published between 1996 and 2007. Even though this dataset has been used to train extractive summarization systems (Durrett et al., 2016; Hong & Nenkova, 2014; Li et al., 2016) or closely-related models for predicting the importance of a phrase in an article (Yang & Nenkova, 2014; Nye & Nenkova,

---

[1] http://duc.nist.gov/

2015; Hong et al., 2015), we are the first group to run an end-to-end abstractive summarization model on the article-abstract pairs of this dataset. While CNN/Daily Mail summaries have a similar wording to their corresponding articles, NYT abstracts are more varied, are shorter and can use a higher level of abstraction and paraphrase. Because of these differences, these two formats are a good complement to each other for abstractive summarization models. We describe the dataset preprocessing and pointer supervision in Section A of the Appendix.

# 6 RESULTS

## 6.1 EXPERIMENTS

**Setup**: We evaluate the intra-decoder attention mechanism and the mixed-objective learning by running the following experiments on both datasets. We first run maximum-likelihood (ML) training with and without intra-decoder attention (removing $c_t^d$ from Equations 9 and 11 to disable intra-attention) and select the best performing architecture. Next, we initialize our model with the best ML parameters and we compare reinforcement learning (RL) with our mixed-objective learning (ML+RL), following our objective functions in Equation 15 and 16. The hyperparameters and other implementation details are described in the Appendix.

**ROUGE metrics and options**: We report the full-length F-1 score of the ROUGE-1, ROUGE-2 and ROUGE-L metrics with the Porter stemmer option. For RL and ML+RL training, we use the ROUGE-L score as a reinforcement reward. We also tried ROUGE-2 but we found that it created summaries that almost always reached the maximum length, often ending sentences abruptly.

## 6.2 QUANTITATIVE ANALYSIS

Our results for the CNN/Daily Mail dataset are shown in Table 1, and for the NYT dataset in Table 2. We observe that the intra-decoder attention function helps our model achieve better ROUGE scores on the CNN/Daily Mail but not on the NYT dataset.

Further analysis on the CNN/Daily Mail test set shows that intra-attention increases the ROUGE-1 score of examples with a long ground truth summary, while decreasing the score of shorter summaries, as illustrated in Figure 2. This confirms our assumption that intra-attention improves performance on longer output sequences, and explains why intra-attention doesnt improve performance on the NYT dataset, which has shorter summaries on average.

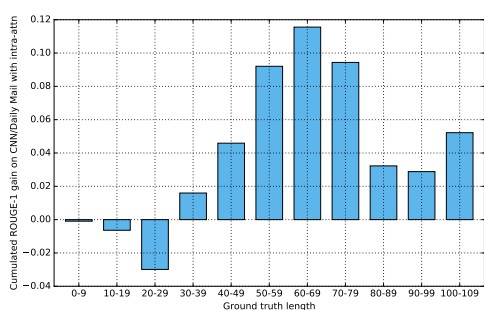

Figure 2: Cumulated ROUGE-1 relative improvement obtained by adding intra-attention to the ML model on the CNN/Daily Mail dataset.

In addition, we can see that on all datasets, both the RL and ML+RL models obtain much higher scores than the ML model. In particular, these methods clearly surpass the state-of-the-art model from Nallapati et al. (2016) on the CNN/Daily Mail dataset, as well as the lead-3 extractive baseline (taking the first 3 sentences of the article as the summary) and the SummaRuNNer extractive model (Nallapati et al., 2017).

See et al. (2017) also reported their results on a closely-related abstractive model the CNN/DailyMail but used a different dataset preprocessing pipeline, which makes direct comparison with our numbers difficult. However, their best model has lower ROUGE scores than their lead-3 baseline, while our ML+RL model beats the lead-3 baseline as shown in Table 1. Thus, we conclude that our mixed-objective model obtains a higher ROUGE performance than theirs.

We also compare our model against extractive baselines (either lead sentences or lead words) and the extractive summarization model built by Durrett et al. (2016), which was trained using a smaller version of the NYT dataset that is 6 times smaller than ours but contains longer summaries. We trained our ML+RL model on their dataset and show the results on Table 4. Similarly to Durrett et al. (2016), we report the limited-length ROUGE recall scores instead of full-length F-scores. For

| Model | ROUGE-1 | ROUGE-2 | ROUGE-L |
|---|---|---|---|
| Lead-3 (Nallapati et al., 2017) | 39.2 | 15.7 | 35.5 |
| SummaRuNNer (Nallapati et al., 2017) | 39.6 | 16.2 | 35.3 |
| words-lvt2k-temp-att (Nallapati et al., 2016) | 35.46 | 13.30 | 32.65 |
| ML, no intra-attention, no trigram avoidance | 35.15 | 13.28 | 32.13 |
| ML, no intra-attention | 37.86 | 14.69 | 34.99 |
| ML, with intra-attention | 38.30 | 14.81 | 35.49 |
| RL, with intra-attention | **41.16** | 15.75 | **39.08** |
| ML+RL, with intra-attention | 39.87 | **15.82** | 36.90 |

Table 1: Quantitative results for various models on the CNN/Daily Mail test dataset

| Model | ROUGE-1 | ROUGE-2 | ROUGE-L |
|---|---|---|---|
| ML, no intra-attention, no trigram avoidance | 42.85 | 26.22 | 39.09 |
| ML, no intra-attention | 44.26 | 27.43 | 40.41 |
| ML, with intra-attention | 43.86 | 27.10 | 40.11 |
| RL, no intra-attention | **47.22** | 30.51 | **43.27** |
| ML+RL, no intra-attention | 47.03 | **30.72** | 43.10 |

Table 2: Quantitative results for various models on the New York Times test dataset

| |
|---|
| **Source document** |
| Jenson Button was denied his 100th race for McLaren after an ERS prevented him from making it to the start-line. It capped a miserable weekend for the Briton; his time in Bahrain plagued by reliability issues. Button spent much of the race on Twitter delivering his verdict as the action unfolded. 'Kimi is the man to watch,' and 'loving the sparks', were among his pearls of wisdom, but the tweet which courted the most attention was a rather mischievous one: 'Ooh is Lewis backing his team mate into Vettel?' he quizzed after Rosberg accused Hamilton of pulling off such a manoeuvre in China. Jenson Button waves to the crowd ahead of the Bahrain Grand Prix which he failed to start Perhaps a career in the media beckons... Lewis Hamilton has out-qualified and finished ahead of Nico Rosberg at every race this season. Indeed Rosberg has now beaten his Mercedes team-mate only once in the 11 races since the pair infamously collided in Belgium last year. Hamilton secured the 36th win of his career in Bahrain and his 21st from pole position. Only Michael Schumacher (40), Ayrton Senna (29) and Sebastian Vettel (27) have more. *(...)* |
| **Ground truth summary** |
| Button denied 100th race start for McLaren after ERS failure. Button then spent much of the Bahrain Grand Prix on Twitter delivering his verdict on the action as it unfolded. Lewis Hamilton has out-qualified and finished ahead of Mercedes team-mate Nico Rosberg at every race this season. Bernie Ecclestone confirms F1 will make its bow in Azerbaijan next season. |
| **ML, with intra-attention (ROUGE-1 41.58)** |
| Button was denied his 100th race for McLaren. ERS prevented him from making it to the start-line. The Briton. He quizzed after Nico Rosberg accused Lewis Hamilton of pulling off such a manoeuvre in China. Button has been in Azerbaijan for the first time since 2013. |
| **RL, with intra-attention (ROUGE-1 50.00)** |
| Button was denied his 100th race for McLaren after an ERS prevented him from making it to the start-line. It capped a miserable weekend for the Briton. Button has out-qualified. Finished ahead of Nico Rosberg at Bahrain. Lewis Hamilton has. In 11 races. . The race. To lead 2,000 laps. . In. . . And. . |
| **ML+RL, with intra-attention (ROUGE-1 44.00)** |
| Button was denied his 100th race for McLaren. The ERS prevented him from making it to the start-line. Button was his team mate in the 11 races in Bahrain. He quizzed after Nico Rosberg accused Lewis Hamilton of pulling off such a manoeuvre in China. |

Table 3: Example from the CNN/Daily Mail test dataset showing the outputs of our three best models after de-tokenization, re-capitalization, replacing anonymized entities, and replacing numbers. The ROUGE score corresponds to the specific example.

| Model | R-1 | R-2 |
|---|---|---|
| First sentences | 28.6 | 17.3 |
| First $k$ words | 35.7 | 21.6 |
| Full (Durrett et al., 2016) | 42.2 | 24.9 |
| ML+RL, with intra-attn | **42.94** | **26.02** |

Table 4: Comparison of ROUGE recall scores for lead baselines, the extractive model of Durrett et al. (2016) and our model on their NYT dataset splits.

| Model | Readability | Relevance | Perplexity |
|---|---|---|---|
| ML | 6.76 | 7.14 | **84.46** |
| RL | 4.18 | 6.32 | 16417.68 |
| ML+RL | **7.04** | **7.45** | 121.07 |

Table 5: Comparison of human readability scores on a random subset of the CNN/Daily Mail test dataset. All models are with intra-decoder attention.

each example, we limit the generated summary length or the baseline length to the ground truth summary length. Our results show that our mixed-objective model has higher ROUGE scores than their extractive model and the extractive baselines.

## 6.3 QUALITATIVE ANALYSIS

We perform human evaluation to ensure that our increase in ROUGE scores is also followed by an increase in human readability and quality. In particular, we want to know whether the ML+RL training objective did improve readability compared to RL.

**Evaluation setup**: To perform this evaluation, we randomly select 100 test examples from the CNN/Daily Mail dataset. For each example, we show the original article, the ground truth summary as well as summaries generated by different models side by side to a human evaluator. The human evaluator does not know which summaries come from which model or which one is the ground truth. Two scores from 1 to 10 are then assigned to each summary, one for relevance (how well does the summary capture the important parts of the article) and one for readability (how well-written the summary is). Each summary is rated by 5 different human evaluators on Amazon Mechanical Turk and the results are averaged across all examples and evaluators.

**Results**: Our human evaluation results are shown in Table 5. Even though RL has the highest ROUGE-1 and ROUGE-L scores, it produces the least readable summaries among our experiments. The most common readability issue observed in our RL results, as shown in the example of Table 3, is the presence of short and truncated sentences towards the end of sequences. This confirms that optimizing for single discrete evaluation metric such as ROUGE with RL can be detrimental to the model quality. On the other hand, our RL+ML summaries obtain the highest readability and relevance scores among our models, hence solving the readability issues of the RL model while also having a higher ROUGE score than ML. This shows the value of the RL+ML training method.

We also report perplexity scores in Table 5. Even though the ML model has the lowest perplexity, it doesn't have the highest readability. This indicate that perplexity measurements cannot replace human judgment for readability evaluation.

## 7 CONCLUSION

We presented a new model and training procedure that obtains state-of-the-art results in text summarization for the CNN/Daily Mail, improves the readability of the generated summaries and is better suited to long output sequences. We also run our abstractive model on the NYT dataset for the first time. We saw that despite their common use for evaluation, ROUGE scores have their shortcomings and should not be the only metric to optimize on summarization model for long sequences. Our intra-attention decoder and combined training objective could be applied to other sequence-to-sequence tasks with long inputs and outputs, which is an interesting direction for further research.

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

# A  NYT DATASET

## A.1  PREPROCESSING

We remove all documents that do not have a full article text, abstract or headline. We concatenate the headline, byline and full article text, separated by special tokens, to produce a single input sequence for each example. We tokenize the input and abstract pairs with the Stanford tokenizer (Manning et al., 2014). We convert all tokens to lower-case and replace all numbers with "0", remove "(s)" and "(m)" marks in the abstracts and all occurrences of the following words, singular or plural, if they are surrounded by semicolons or at the end of the abstract: "photo", "graph", "chart", "map", "table" and "drawing". Since the NYT abstracts almost never contain periods, we consider them multi-sentence summaries if we split sentences based on semicolons. This allows us to make the summary format and evaluation procedure similar to the CNN/Daily Mail dataset. These pre-processing steps give us an average of 549 input tokens and 40 output tokens per example, after limiting the input and output lengths to 800 and 100 tokens.

## A.2  DATASET SPLITS

We created our own training, validation, and testing splits for this dataset. Instead of producing random splits, we sorted the documents by their publication date in chronological order and used the first 90% (589,284 examples) for training, the next 5% (32,736) for validation, and the remaining 5% (32,739) for testing. This makes our dataset splits easily reproducible and follows the intuition that if used in a production environment, such a summarization model would be used on recent articles rather than random ones.

## A.3  POINTER SUPERVISION

We run each input and abstract sequence through the Stanford named entity recognizer (NER) (Manning et al., 2014). For all named entity tokens in the abstract if the type "PERSON", "LOCATION", "ORGANIZATION" or "MISC", we find their first occurrence in the input sequence. We use this information to supervise $p(u_t)$ (Equation 11) and $\alpha_{ti}^e$ (Equation 4) during training. Note that the NER tagger is only used to create the dataset and is no longer needed during testing, thus we're not adding any dependencies to our model. We also add pointer supervision for out-of-vocabulary output tokens if they are present in the input.

# B  HYPERPARAMETERS AND IMPLEMENTATION DETAILS

For ML training, we use the teacher forcing algorithm with the only difference that at each decoding step, we choose with a 25% probability the previously generated token instead of the ground-truth token as the decoder input token $y_{t-1}$, which reduces exposure bias (Venkatraman et al., 2015). We use a $\gamma = 0.9984$ for the ML+RL loss function.

We use two 200-dimensional LSTMs for the bidirectional encoder and one 400-dimensional LSTM for the decoder. We limit the input vocabulary size to 150,000 tokens, and the output vocabulary

to 50,000 tokens by selecting the most frequent tokens in the training set. Input word embeddings are 100-dimensional and are initialized with GloVe (Pennington et al., 2014). Based on these dimensions and sizes, our final model has 16.9M trainable parameters, 15M of which are word embeddings.

We train all our models with Adam (Kingma & Ba, 2014) with a batch size of 50 and a learning rate $\alpha$ of 0.001 for ML training and 0.0001 for RL and ML+RL training. At test time, we use beam search of width 5 on all our models to generate our final predictions.

