# OpenReview forum: "A Deep Reinforced Model for Abstractive Summarization"
_ICLR.cc/2018/Conference — Accept (Poster)_

### Official Review · AnonReviewer1 · 2017-11-27
**Strong empirical contribution**

**Rating:** 8
**Confidence:** 3

**Review:**

The paper proposes a model for abstractive document summarization using a self-critical policy gradient training algorithm, which is mixed with maximum likelihood objective. The Seq2seq architecture incorporates both intra-temporal and intra-decoder attention, and a pointer copying mechanism. A hard constraint is imposed during decoding to avoid trigram repetition. Most of the modelling ideas already exists, but this paper show how they can be applied as a strong summarization model.

The approach obtains strong results on the CNN/Daily Mail and NYT datasets. Results show that intra-attention improves performance for only one of the datasets. RL results are reported with only the best-performing attention setup for each dataset. My concern with that is that the authors might be using the test set for model selection; It is not a priori clear that the setup that works better for ML should also be better for RL, especially as it is not the same across datasets. So I suggest that results for RL should be reported with and without intra-attention on both datasets, at least on the validation set.

It is shown that intra-decoder attention decoder improves performance on longer sentences. It would be interesting to see more analysis on this, especially analyzing what the mechanism is attending to, as it is less clear what its interpretation should be than for intra-temporal attention. Further ablations such as the effect of the trigram repetition constraint will also help to analyse the contribution of different modelling choices to the performance.

For the mixed decoding objective, how is the mixing weight chosen and what is its effect on performance? If it is purely a scaling factor, how is the scale quantified? It is claimed that readability correlates with perplexity, so it would be interesting to see perplexity results for the models. The lack of correlation between automatic and human evaluation raises interesting questions about the evaluation of abstractive summarization that should be investigated further in future work.

This is a strong paper that presents a significant improvement in document summarization.

---

### Official Review · AnonReviewer2 · 2017-11-27
**Well-motivated, with clear experimental results and thorough evaluations.**

**Rating:** 7
**Confidence:** 5

**Review:**

This is a very clearly written paper, and a pleasure to read.

It combines some mechanisms known from previous work for summarization (intra-temporal attention; pointing mechanism with a switch) with novel architecture design components (intra-decoder attention), as well as a new training objective drawn from work from reinforcement learning, which directly optimizes ROUGE-L. The model is trained by a policy gradient algorithm.

While the new mechanisms are simple variants of what is taken from existing work, the entire combination is well tested in the experiments. ROUGE results are reported for the full hybrid RL+ML model, as well as various versions that drop each of the new components (RL training; intra-attention). The best method finally outperforms the lea-3d baseline for summarization. What makes this paper more compelling is that they compared against a recent extractive method (Durret et al., 2016), and the fact that they also performed human readability and relevance assessments to demonstrate that their ML+RL model doesn't merely over-optimize on ROUGE. It was a nice result that only optimizing ROUGE directly leads to lower human evaluation scores, despite the fact that that model achieves the best ROUGE-1 and ROUGE-L performance on CNN/Daily Mail.

Some minor points that I wonder about:
 - The heuristic against repeating trigrams seems quite crude. Is there a more sophisticated method that can avoid redundancy without this heuristic?
 - What about a reward based on a general language model, rather than one that relies on L_{ml} in Equation (14)? If the LM part really is to model grammaticality and coherence, a general LM might be suitable as well.
 - Why does ROUGE-L seem to work better than ROUGE-1 or ROUGE-2 as the reward? Do you have any insights are speculations regarding this?

---

### Official Review · AnonReviewer3 · 2017-11-28
**Good incremental research but not exciting**

**Rating:** 6
**Confidence:** 4

**Review:**

The paper is generally well-written and the intuition is very clear. It combines the advanced attention mechanism, pointer networks and REINFORCE learning signal to train a sequence-to-sequence model for text summarization. The experimental results show that the model is able to achieve the state-of-the-art performance on CNN/Daily Main and New York Times datasets. It is a good incremental research, but the downside of this paper is lack of innovations since most of the methods proposed in this paper are not new to us.

I would like to see the model ablation w.r.t. repetition avoidance trick by muting the second trigram at test time. Intuitively, if the repetition issue is prominent to having decent summarization performance, it might affect our judgement on the significance of using intra-attention or combined RL signal.
Another thought on this: is it possible to integrate the trigram occurrence with summarization reward? so that the recurrent neural networks with attention could capture the learning signal to avoid the repetition issue and the heuristic function in the test time can be removed.

In addition, as the encoder-decoder structure gradually becomes the standard choice of sequence prediction, I would suggest the authors to add the sum of parameters into model ablation for reference.

Suggested References:
Bahdanau et al. (2016) An Actor-critic Algorithm for Sequence Prediction. (actor-critic on machine translation)
Miao & Blunsom (2016) Language as a Latent Variable: Discrete Generative Models for Sentence Compression. (mixture pointer mechanism + REINFORCE)

---

### Public Comment · (anonymous) · 2017-12-07
**Request for code**

Hi! I think your paper is very very instructive. Can you share the code with me? Email Address:zhu1qingqing@gmail.com                  Thank you!

---

### Comment · Area_Chair · 2017-12-27
**Discussion**

Authors,

Please respond to the reviewers if you have any rebuttal points. While scores are positive, it is helpful to have these points resolved.

---

### Author Response · Authors · 2018-01-04
**Revisions and updated PDF**

Following the helpful comments and feedback from all reviewers, we updated our paper submission with the following changes:
- Add the number of parameters of our model
- Add model ablation results with respect to the trigram avoidance trick on the CNN/Daily Mail and New York Times datasets
- Add perplexity scores and compare them with human evaluation results
- Add Bahdanau et al. (2016) and Miao & Blunsom (2016) citations
- Other minor fixes in citations

---

### Decision · Program_Chairs · 2018-01-29
**ICLR 2018 Conference Acceptance Decision**

**Decision:**

Accept (Poster)

**Comment:**

This work extends upon recent ideas to build a complete summarization system using clever attention, copying, and RL training. Reviewers like the work but have some criticisms. Particularly in terms of its originality and potential significance  noting "It is a good incremental research, but the downside of this paper is lack of innovations since most of the methods proposed in this paper are not new to us.". Still reviewers note the experimental results are of high quality performing excellent on several datasets and building "a strong summarization model." Furthermore the model is extensively tested including in "human readability and relevance assessments ".  The work itself is well written and clear.